# Molecular Functions of Thyroid Hormone Signaling in Regulation of Cancer Progression and Anti-Apoptosis

**DOI:** 10.3390/ijms20204986

**Published:** 2019-10-09

**Authors:** Yu-Chin Liu, Chau-Ting Yeh, Kwang-Huei Lin

**Affiliations:** 1Department of Biochemistry, College of Medicine, Chang-Gung University, Taoyuan 333, Taiwan; k1506820@gmail.com; 2Department of Biomedical Sciences, College of Medicine, Chang-Gung University, Taoyuan 333, Taiwan; 3Liver Research Center, Chang Gung Memorial Hospital, Taoyuan 333, Taiwan; chauting@adm.cgmh.org.tw; 4Research Center for Chinese Herbal Medicine, College of Human Ecology, Chang Gung University of Science and Technology, Taoyuan 333, Taiwan

**Keywords:** thyroid hormone, thyroid hormone receptor, 3,3,5-triiodo-L-thyronine (T_3_), L-thyroxine (T_4_), cancer proliferation

## Abstract

Several physiological processes, including cellular growth, embryonic development, differentiation, metabolism and proliferation, are modulated by genomic and nongenomic actions of thyroid hormones (TH). Several intracellular and extracellular candidate proteins are regulated by THs. 3,3,5-Triiodo-L-thyronine (T_3_) can interact with nuclear thyroid hormone receptors (TR) to modulate transcriptional activities via thyroid hormone response elements (TRE) in the regulatory regions of target genes or bind receptor molecules showing no structural homology to TRs, such as the cell surface receptor site on integrin αvβ3. Additionally, L-thyroxine (T_4_) binding to integrin αvβ3 is reported to induce gene expression through initiating non-genomic actions, further influencing angiogenesis and cell proliferation. Notably, thyroid hormones not only regulate the physiological processes of normal cells but also stimulate cancer cell proliferation via dysregulation of molecular and signaling pathways. Clinical hypothyroidism is associated with delayed cancer growth. Conversely, hyperthyroidism is correlated with cancer prevalence in various tumor types, including breast, thyroid, lung, brain, liver and colorectal cancer. In specific types of cancer, both nuclear thyroid hormone receptor isoforms and those on the extracellular domain of integrin αvβ3 are high risk factors and considered potential therapeutic targets. In addition, thyroid hormone analogs showing substantial thyromimetic activity, including triiodothyroacetic acid (Triac), an acetic acid metabolite of T_3_, and tetraiodothyroacetic acid (Tetrac), a derivative of T_4_, have been shown to reduce risk of cancer progression, enhance therapeutic effects and suppress cancer recurrence. Here, we have reviewed recent studies focusing on the roles of THs and TRs in five cancer types and further discussed the potential therapeutic applications and underlying molecular mechanisms of THs.

## 1. Introduction

Thyroid hormone (TH) in adults is necessary for the regulation of multiple physiological effects, such as cell growth, structure, and metabolism [1]. The main thyroid hormones produced by the thyroid gland are thyroxine (T_4_), 3,5,3′-triiodothyronine (T_3_), and reverse 3,5,3′-triiodothyronine (rT_3_), which are controlled by thyroid-stimulating hormone (TSH). Under physiological conditions, both T_4_ and T_3_ are secreted into the bloodstream by the thyroid gland [2]. THs circulating in the body exert metabolic effects on multiple organs, including heart, bone, brain, liver, thyroid, kidney and skeletal muscle [2]. The actions of thyroid hormone are classified into two main mechanisms: (1) A non-genomic effect initiated at the plasma membrane that regulates downstream gene expression via integrin αvβ3; and (2) transcriptional activity induced by interactions with nuclear thyroid hormone receptor proteins and further binding to thyroid hormone response elements of specific downstream genes. The αvβ3 isoform of integrin is a heterodimeric structure at the plasma membrane capable of interacting with a large number of extracellular matrix (ECM) proteins as ligands for activating downstream signal pathways [3]. In addition, the protein structure of thyroid hormone receptor is similar to nucleus receptor superfamily and acts as a sequence-specific ligand-dependent transcription factor that mediates several downstream effects of THs on activation or repression of target genes [4]. The above actions generated through either non-genomic or genomic effects overlap in the nucleus. Thyroid hormone activity is beneficial for normal cell development. However, when both the levels of THs and thyroid hormone receptors in the body are out of control, it causes multiple diseases, including cardiovascular disease, diabetes mellitus and chronic liver disease [5]. Earlier studies by our group and other investigators conducted to clarify the significance of thyroid hormone in cells and tissues have revealed activity in regulation of proliferation of both tumor and nonmalignant cells. The current review focuses on the potential association between thyroid hormones and progression of different cancer types.

## 2. Thyroid Hormone Effects via Interactions with the Thyroid Hormone Receptor

### 2.1. Thyroid Hormone

The thyroid hormone system starts from the hypothalamus, where thyrotropin-releasing hormone (TRH) is synthesized and released from the periventricular nucleus (PVN). TRH binding to its receptor on the thyrotroph of the anterior pituitary gland stimulates proliferation, synthesis and secretion of thyroid stimulating hormone (TSH). TSH subsequently interacts with the TSH receptor (TSHR) on individual thyroid follicular cells of the thyroid gland to stimulate synthesis and release of thyroid hormones L-thyroxine (T_4_) and 3,3,5-triiodo-L-thyronine (T_3_) [6]. In addition, thyroglobulin, a dimeric protein, is synthesized in the rough endoplasmic reticulum of thyroid follicular cells and secreted to enter the follicular colloid via exocytosis. Simultaneously, iodide (I^-^) is transported to thyroid follicular cells via sodium-iodide (Na/I) symporter pump activity and enters thyroid follicular cells from the cytoplasm in a pendrin-dependent manner. One of the enzymes in the follicular colloid, thyroid peroxidase, catalyzes iodide (I^−^) oxidization to iodine (I^0^). Iodine (I^0^) iodinates thyroglobulin and conjugates with the protein chain of tyrosyl residues. Subsequently, thyroglobulin re-enters thyroid follicular cells via endocytosis and undergoes proteolysis via actions of various proteases to liberate thyroxine (T_4_) and 3,3,5-triiodo-L-thyronine (T_3_). Efflux of T_4_ and T_3_ from thyroid follicular cells to various target cells is achieved through specific membrane transporter proteins [7], such as monocarboxylate transporter (MCT) 8 and 10 [8,9], the organic anion transporter protein-1c1 (OATP1c1), and nonspecific L-type amino acid transporters 1 and 2 (LAT1, LAT2) [10]. However, earlier studies indicate that euthyroid status in blood with circulating T_4_ and T_3_ is controlled by a negative feedback loop mediated by the hypothalamus-pituitary-thyroid (HPT) set axis [11].

### 2.2. Thyroid Hormone Receptor

Circulating THs interact with thyroid hormone receptors to promote downstream signaling pathways and activate transcription factors. Thyroid hormone receptors (TR) including TRα and TRβ contain several domains, specifically, amino terminal A/B that may function as a gene enhancer, DNA-binding domain (DBD), hinge region containing the nuclear localization signal and carboxy-terminal ligand-binding domain that binds T_3_ (Figure 1). These receptors display protein structures similar to most nuclear receptors and each domain performs specific functions [4,12,13,14,15]. In particular, the amino-terminal A/B of thyroid hormone receptors (TRβ1 and TRβ2) is generated from a signaling gene via alternative splicing or usage of alternative promoters [16]. The four major TR isoforms, TRα1, TRα2, TRβ1, and TRβ2, are produced by *c-erbAα* and *c-erbAβ* genes. Their human homologs are designated THRA and THRB. The *c-erbAα* gene located on chromosome 17 encodes two different TRα isoforms. One is functional TH-binding TRα1 and the other is a dominant-negative splice variant, TRα2, lacking TH binding activity [17]. TRα2 is unique in consideration of its lack of binding to THs while interacting with DNA, and its precise function is unclear at present. The *c-erbAβ* gene located on chromosome 3 encodes two isoforms, TRβ1 and TRβ2, that participate in TH binding and are widely distributed in a tissue-specific manner. TRα1 and TRα2 are expressed in the kidney, skeletal muscle, lungs, heart, and testes, with particularly high levels detected in the brain [18]. TRβ1 expression is significant in brain, thyroid, liver, and kidney while the TRβ2 isoform is specifically expressed in the anterior pituitary, hypothalamus, and developing brain [12,19,20,21] (Figure 1).

Interestingly, TRα1, TRα2 and TRβ1 are overexpressed in various tissues in the human body excepting the liver, the major TH target organ [17]. Additionally, TRs regulate transcriptional activity through associating with other nuclear receptors such as retinoid X receptor (RXR), retinoic acid receptor subtypes, and vitamin D receptor (VDR), which are homo or heterodimers (Figure 2A). TRs/RXR belong to non-permissive heterodimers that can be activated transcriptionally by TRs ligand but not by RXR ligand alone (Figure 2A). In particular, RXR forms a heterodimer with TRs that influence downstream target gene expression by binding to specific DNA sequences located in regulatory regions known as thyroid hormone elements (TRE) [22,23,24]. TREs within the promoter region contain individual DNA sequences (A/G)GGT(C/A/G)A, designated “half-sites”, that are recognized by TRs. The half-site sequences in TREs incorporate palindromic (Pal), direct repeat (DR) and inverted repeat arrangements (IP) [15] (Figure 2B). In the absence of thyroid hormone in cells, TRs interacting with TREs do not influence gene expression. TRs enter the nucleus and bind to DNA until arrival of the thyroid hormone.

### 2.3. Nuclear Transcriptional Activity of Thyroid Hormone

Several recent studies have focused on transcriptional activation induced by TR binding to positive TRE sites. However, TRs play a dual modulatory role and can also repress gene expression in a ligand-dependent manner. TRs bind to their respective TREs as monomers, homodimers or heterodimers with retinoid X receptors (RXR). Heterodimers of TR/RXR contain LBDs that interact with T_3_. Additionally, DBD displays high affinity for DNA sequences of TREs [25]. TRs can directly or indirectly associate with different molecular proteins (transcription factors, coactivators, transcription intermediary factors (TIF) and corepressors) to influence downstream target gene expression. Biochemical analyses have demonstrated that nuclear receptor corepressor (NCoR) and homolog, silencing mediator of retinoic and thyroid receptor (SMRT), are strongly associated with unliganded TR [26]. A number of studies suggest that the corepressor interacts with the TR homodimer but not monomer on DNA. Recruitment of NCoR and SMRT to the promoter via association with DBD of TR/RXR leads to strong repression of basal promoter activity of target genes. NCoR and SMRT are structurally and functionally similar. Both proteins contain three repressor domains (RD1, RD2, and RD3) and two receptor-interacting domains (RID). RD1 interacts with TBL1 and mSin3, which recruit class I deacetylases (HDAC1 and HDAC2). HDAC3 interacts directly with RD2. Moreover, class II deacetylases (HDAC4/5 and HDAC7) bind RD3 as a mediator for repressing downstream genes transcription. In other words, RD1, RD2 and RD3 domains of corepressors interact with different types of deacetylase and other proteins to form a large repressor complex that suppresses target gene transcription [27] (Figure 3A). Recent studies have shown strong links among histone acetylation, chromatin remodeling and gene regulation activities [28,29]. As specified above, NCoR and SMRT may function as corepressors via histone deacetylase activity for complex-mediated chromatin remodeling. HDACs and histone binding proteins RbAp46/48 associate with the homologs mSin3A and mSin3B. HDACs are recruited to target genes by associating with Sin3 protein that interacts with sequence-specific DNA binding factors [30] (Figure 3B). The mSin3-HDAC complex is highly abundant and stable, facilitating binding and recruitment by the nuclear receptor repressors NCoR and SMRT [31,32]. Specifically, HDACs interact with mSin3 for TR/RXR-mediated repression. The TR/RXR/mSin3 complexes are indirectly mediated by NCoR and SMRT, which function to link receptors to mSin3-HDAC complexes. NCoR and SMRT corepressors are reported to recruit class I deacetylases after interactions with adaptor mSin3 protein. However, biochemical research to date has failed to detect NCoR or SMRT in mSin3-HDAC complexes. Other class II histone deacetylases (HDAC4, HDAC5 and HDAC7) have been identified that associate with SMRT interacting-proteins and repress gene transcription (Figure 3B). Therefore, a single corepressor can mediate downstream target gene expression via class I HDAC complexes in a Sin3A-dependent manner or class II HDAC complexes in a Sin3A-independent manner. Furthermore, a novel SMRT-containing cellular complex incorporating HDAC3 and transducing beta-like protein 1 (TBL1), a protein that interacts with histone H3, has been identified (Figure 3B). In vivo, TBL1 is involved in a repressor complex that repression of gene transcription through is bridged to HDAC3 and further interactions with SMRT, and can potentiate repression via TR [33]. However, under TH presence conditions, the TR conformation is altered to allow dissociation of corepressors (NCoR or SMRT), subsequent recruitment of transcriptional coactivators and induction of target gene transcription. Multiprotein complexes designated ‘TRAP’ interact with TRs [34,35] (Figure 3B). Biochemical analyses have revealed binding of a series of proteins to TRs, the most abundant of those proteins are molecularly distinguished with a molecular mass of 140 or 160 kDa, designated p140 and p160 [34]. Steroid receptor coactivator 1 (SRC-1) is the first p160 family member identified as a coactivator and interacts with several nuclear receptors via C-terminal activation function-2 (AF-2) in the ligand binding domain (LBD), one of which is TR [36,37]. SRC-1 has been shown to enhance the transcriptional activity of a number of transcription factors [38]. SRC-1 and other SRC families of displaying coactivator structures contain a basic-helix-loop-helix (bHLH) and single-minded (PAS) domain, which interact with intermolecular or intramolecular substrates [39]. In addition, SRC-1 has a nuclear receptor interacting domain incorporating three LXXLL amino acid motifs, which are not only required for nuclear receptor binding but also recruit specificity to other proteins [40,41,42]. Another significant finding is that SRC-1 coactivators contain two activation domains in the C-terminal region, denoted AD1 and AD2. The stronger transactivation domain, AD1, interacts with co-integrator CREB binding protein (CBP) [43] and another weaker activation domain in the far C-terminal region while AD2 interacts with an arginine methyltransferase, CARM1 (Figure 3A). A number of studies suggest that the SRC family acts as a platform for recruitment of other proteins. Histone acetyltransferase (HAT) activity is possessed by CREB-binding protein (CBP)/p300 and p300/CBP-associated factor (p/CAF), which modulate chromatin remodeling [44,45] via acetylation of histone H3 and H4 (Figure 3A,B). SRC-1 also plays a potential adapter role and is capable of interacting with specific basal transcription factors, such as TATA binding factor (TBP) and transcription factor IIB [46]. Multiprotein complexes known as TR-associated proteins (TRAP) have been identified that interact with TRs [34,35]. These complexes can indirectly interact with TRs in response to ligand binding via a receptor-interacting LXXLL motif in the p160 coactivator through TRAP 220 [47]. In a cell-free transcription system with chromatin templates, ligand activity of TRs is enhanced by the TRAP complex, as determined from studies on SRC-1 [48] (Figure 3B). Several investigations to date indicate that ligand-dependent transcriptional activity of TRs requires recruitment of p160-TRAP coactivator complexes but the function association between TRAP complexes and the p160/CBP/PCAF system is unclear. For example, both TRAPs and p160/CBP/PCAF interact with the same region of TRs but do not bind simultaneously, and the molecule that binds first is yet to be established.

### 2.4. Non-Genomic Actions of Thyroid Hormone

The preliminary studies have revealed in the mitochondria and cytoskeleton that action of thyroid hormone that is not primarily involved in nuclear activities [49,50]. These actions are rapid in contrast to transcription and translation processes that occur over minutes or hours and are not exerted through gene transcription or protein synthesis, leading to coining of the term “non-genomic effects”. Non-genomic activity of the thyroid hormone in enucleate cells, plasma membrane and other cell fractions in vitro has been identified. According to these earlier studies, thyroid hormone receptors on integrin αvβ3 are not homologous to nuclear thyroid hormone receptors. THs affect multiple physiological activities within the cell via interactions with integrin αvβ3 [51]. The integrin family has 24 structural proteins in the plasma membrane that essentially regulate cell–cell and cell–extracellular matrix (ECM) protein interactions [52]. αvβ3, one of the isoforms of integrin, is heterodimeric in structure and contains an Arg–Gly–Asp (RGD) recognition specific binding site for ECM proteins, such as osteopontin, fibronectin and vitronectin, for activation of intracellular signaling [52]. In other words, thyroid hormones bind the receptor near the RGD site of integrin αvβ3 that serves as a recognition and binding motif for ECM proteins [53]. Upon interaction of integrin αvβ3 with THs (T_3_ and T_4_), mitogen-activated protein kinase/extracellular signal-regulated kinase (MAPK/ERK 1/2) pathways are activated that regulate multiple cellular physiological processes [51] (Figure 4A). Integrin αvβ3 contains two binding domains: (1) S1 that specifically recognizes T_3_ and activates the phosphatidylinositol 3-kinase (PI3K)/Akt/protein kinase B (PKB) pathway via stimulation of Src kinase (Figure 4B) and (2) S2 that binds both T_3_ and T_4_, leading to regulation of MAPK/ERK1/2 (Figure 4A). T_4_ has high affinity for the S2 domain while both S1 and S2 domains interact with T_3_. In addition, the two binding domains mediate distinct downstream effects. For instance, Src kinase and PI3K pathways are activated by T_3_ binding to the S1 domain, leading to direct trafficking of TRα1 from the cytoplasm to the nucleus and transcriptional activity of the target gene, *HIF1A* (Figure 4B). The S2 domain stimulates activation of MAPK/ERK1/2 via phospholipase C (PLC) and protein kinase Cα (PKCα), promoting phosphorylation of nucleoproteins TRP1, ERM, STAT1 and p35 [54,55] and modulation of intracellular protein trafficking of estrogen receptor α (ERα) and nuclear uptake of TRβ from the cytoplasm [56,57] (Figure 4A). Notably, estrogen receptor α (ERα) in this pathway is phosphorylated, suggestive of crosslinking between thyroid and steroid hormone pathways [58]. These represent two of the five mechanisms underlying the non-genomic action of thyroid hormone. In the cytoplasm, the PI3K pathway is rapidly activated via T_3_ interactions with TRβ1 and initiates downstream target gene transcription (Figure 4C). Interactions of T_3_-liganded TRβ1 with the regulatory p85α subunit of PI3K that induces downstream AKT phosphorylation lead to subsequent phosphorylation of mTOR and further activation of mTOR-p70S6K along with a series of downstream target genes in the nucleus, such as hypoxia inducible factor-1α (HIF-1α), glucose transporter 1 (GLUT1), platelet-type phosphofructokinase (PFKP) and monocarboxylate transporter 4 (MCT 4) [59,60,61] (Figure 4C). Additionally, at the plasma membrane, T_3_ interacts with truncated TRα1 (p30 TRα1), and binding of PI3K to p85α inactivates transcription of downstream genes. However, the T_3_-liganded TRα1 complex activates a series of signal transduction proteins (PKGII and ERK) and nitric oxide synthase (NOS) (Figure 4D). Additionally, T_3_ interactions with TRβ1 are reported to modulate Na, K-ATPase activity via activation of both MAPK/ERK 1/2 and PI3K pathways (Figure 4C,E). However, only T_3_-liganded TRβ1 stimulates MAPK/ERK 1/2 activity, which activates the sodium proton exchanger (Na^+^/H^+^) in the plasma membrane [57,62] (Figure 4E).

## 3. Functional Significance of Thyroid Hormone and Receptors in Tumors

Under physiological conditions, thyroid hormone receptors control tumor cell proliferation and cancer cell defense pathways [54,59,63]. TRs are reported to exert tumor suppressor effects [64], with 96% nuclear TRβ1 expression detected in the normal epithelium but lower frequency of expression in adenomas (~83%) and cancer (68%), which is significantly lower than that in normal tissue and adenoma). Consistently, another study demonstrated a tumor suppressor role of wild-type TRβ1 in thyroid cancer [65,66,67], similar to that in other cancer types [67,68]. Conversely, abnormal expression or mutation of TRβ1 has been shown to promote carcinogenesis [69]. Interestingly, clinical data on BRCA1-associated breast cancer suggest that TRβ1 expression can extend the overall survival curve. However, wild-type TRα was positively associated with reduced five-year overall survival for five years [70]. Wild-type TRα plays a distinct role in cancer relative to TRβ1, which TRα potentially influences tumorigenesis and hematogenous metastasis via association with *nm23* genes [64,70]. Additionally, TRα binding to T_3_ promotes gastrointestinal cancer development through directly modulating the transcriptional activity of β-catenin and affecting downstream signal transduction [71]. Thyrotropin, THs, integrin αvβ3 and deiodinases are involved in cancer proliferation along with TRs. Earlier studies suggest that altered TH status modulates cancer cell proliferation and tumor growth. For example, low expression of circulating thyrotropin releasing hormone (TRH) is associated with increased risk of lung, colon, prostate, and breast cancer [72]. Hyperthyroidism in rodents stimulates tumor transplant growth and metastasis, and conversely, hypothyroidism suppresses these effects [73]. Furthermore, both breast and prostate cancers at advanced clinical stages display high expression of THs. Spontaneous hypothyroidism may beneficially alter the course and aggressiveness of breast cancer [74]. Clearly, TH expression is correlated with cancer initiation. In the following section, the underlying mechanisms modulating proliferation and metastasis of different cancer types, including thyroid hormone effects, deiodinase activities and thyroid hormone receptors, are discussed (Table 1).

### 3.1. Breast Cancer

Thyroid hormone activity is related to breast cancer. Hypothyroidism is associated with low incidence of breast cancer, and conversely, hyperthyroidism with high incidence and aggressiveness of breast cancer [99]. Statistical studies suggest that high T_3_ levels in women are associated with increased overall risk and occurrence of breast cancer. A study on 1322 peri/postmenopausal women disclosed a significant direct correlation between T_3_ level and breast cancer occurrence [100]. High T_4_ and T_3_ expression in the blood leads to subclinical hyperthyroidism in menopausal breast cancer patients [101]. On the other hand, the most frequent endocrine disorders are encountered in breast cancer. An earlier clinical study showed that among 844 breast cancer patients, 74 were hypothyroid and not only older but also diagnosed at an earlier stage than euthyroid patients [74]. Hypothyroidism is reported to promote breast cancer apoptosis and suppress mammary carcinogenesis through alterations in body composition, including leptin secretion and serum 17β-estradiol (E2) [102]. Additionally, hypothyroidism can decrease inherent drug resistance and induce chemotherapy sensitivity [103]. Evidence of overlap of both non-genomic and genomic actions of thyroid hormones (THs) in breast cancer cells and crosslinking with other hormones, such as estrogens and testosterone, has been uncovered. TH-induced phosphorylation of estrogen receptor α (ERα) at Ser-118 via activation of the MAPK/ERK 1/2 pathway promotes human breast cancer cell proliferation [58,75]. In human breast cell lines MCF-7 and T47D, both T_4_ and T_3_ promote cell proliferation in a dose-dependent manner [104,105,106]. Hyperthyroidism and hypothyroidism are correlated with the incidence of tumor induction and reduction and aggressiveness of breast cancer in female mice [99]. Interestingly, other reports support anticancer activity of THs, for instance, via downregulation of *SMP30* expression and induction of apoptosis in MCF-7 cells through T_3_ [76]. Both T_4_ and T_3_ have been shown to regulate the breast cancer process. T_4_ only appears to interact with cell surface receptors of the hormone whereas T_3_ binds nuclear THRs via genomic activity as well as integrin αvβ3 in the cytoplasm [107]. As stated above, physiological free T_4_ acts as a growth factor that influences cancer progression, anti-apoptosis and endothelial cell migration. The main function of T_3_ is to promote breast cancer cell division [54,79,82,108,109]. *T1* gene overexpression in breast adenocarcinoma is induced by mitogens, serum, specific oncogenes and cytokines [77] and inhibited under conditions of high T_3_ concentrations, leading to reduced breast cancer cell proliferation. In addition, T_3_ suppresses STAT5-mediated regulation of downstream target gene expression by inhibiting STAT5 signaling, which can associate with TRβ1 as a tumor suppressor protein and inhibit mammary hyperplasia development [78]. TRβ1 has been shown to function as a tumor suppressor in a number of cancer types (including breast, lung and thyroid cancer). In a xenograft mouse experiment, injection of MCF-7-Neo cells into athymic mice promoted rapid tumor development. Conversely, when MCF-7 TRβ was injected into athymic mice, tumor growth was inhibited through serum 17β-estradiol (E2) [78]. In nuclei of breast cancer cells, both TRβ and TRα proteins are expressed, with 74% of breast tumors highly expressing TRα1 and 40% TRα2. TRα2 influences prognostic histopathological parameters in breast cancer patients, such as tumor size, axillary lymph node involvement, grading and hormone receptor status, leading to improvement of overall survival [110,111].

### 3.2. Thyroid Cancer

Three morphological subtypes (papillary thyroid carcinoma, follicular thyroid carcinoma (FTC) and anaplastic carcinoma) constitute 90% of all thyroid cancer types [112]. The majority of studies indicate that thyroid cancer is commonly associated with hyperthyroidism [4,113]. T_4_ induces proliferation of most human carcinoma cells (including follicular and papillary thyroid carcinoma cell lines) through binding to cell surface receptors on integrin αvβ3. Interactions of T_4_ with integrin αvβ3 lead to inhibition of p53-dependent apoptosis in tumor cells. Additionally, in differentiated thyroid carcinomas (DTC), TSH act as a growth factor predominantly through interactions with the thyrotropin receptor on papillary and follicular thyroid cancer cells. Thus, lowering or suppression of host TSH with exogenous T_4_ can be applied for standardized primary treatment and long-term management of DTC. On the other hand, T_4_ can also be used to suppress TSH in recurrent disease, supporting a critical role of T_4_ in modulating DTC proliferation and recurrence via influencing TSH [114]. In vitro findings suggest that treatment with external THs enhances proliferation of human papillary and follicular thyroid cancer cells. Specifically, THs interact with plasma membrane integrin αvβ3, activating MAPK/ERK1/2 signaling and promoting papillary and follicular thyroid cancer cell proliferation and anti-apoptosis [79]. Additionally, experiments on a mouse model of thyroid carcinoma primary and metastatic lesions showed activation of PI3K-Akt signaling upon mutation of TRβ. Thyroid hormone β receptor (TRβ)PV/PV mice with knock-in mutant TRβ gene (TRβPV mutant) spontaneously developed thyroid cancer and distant metastasis similar to human follicular thyroid cancer. Furthermore, in these spontaneous thyroid tumors, the ligand binding domain of TR interacted with PI3K regulatory subunit, p85α, to a greater extent than wild-type TRβ. LY-294002 is a PI3K signaling specific inhibitor, which blocks rapamycin-p70(S6K) of Akt mammalian signaling, leading to both increased p27 and decreased cyclin D1, further inhibiting thyroid tumor growth and tumor cell proliferation. LY294002 treatment promotes thyroid tumor apoptosis by increasing caspase-3 expression and reducing phosphorylated BAD and suppresses thyroid cancer cell motility, thus influencing metastatic ability [80]. Data from this study suggest that mutation of TRβ plays a critical role in thyroid cancer development. In another study, a mouse model with dominant-negative mutant thyroid hormone receptor β (denoted PV) was generated. Reduced PI3K activation by T_3_ in Thrb(PV/PV) mice was observed, along with inability to bind to mutant TRβ under conditions of hypothyroidism [66]. The data suggest that thyroid gland carcinogenesis in Thrb(PV/PV) mice is promoted by thyroid hormone via binding to cell surface integrin αvβ3 [66]. Deiodinase type 3 (DIO3, D3) that converts T_4_ to T_3_ is upregulated in the human PTC-derived cell line, K1, by transforming growth factor β1 (TGFβ1). Additionally, treatment with the inhibitors U0126 (ERK pathway) and SB203580 (p38 pathway) led to blockage of the MAPK pathway, and subsequent decrease in DIO3 mRNA and inhibition of DIO3 transcriptional induction via TGFβ1, clearly suggesting that D3 is upregulated via MAPK signaling [115]. The collective findings indicate that D3 expression is positively correlated with thyroid tumor size and disease spread [115].

### 3.3. Lung Cancer

Around 60% of patients diagnosed with lung cancer present an advanced stage of disease that is too late for surgical treatment. Small-cell lung carcinoma is associated with symptoms of hyperthyroidism [116]. Lung cancer patients often present with non-thyroidal illness syndrome (NTIS) or sick euthyroid syndrome, characterized by alterations in circulating TH expression in acute or chronic systemic disease. In NTIS patients, the circulating T_3_ level is decreased and T_4/_T_3_ expression ratio significantly increased along with modest alterations in regulation of rT_3_. Over a six-month observation period, mortality of lung cancer patients with low-level T_3_ was higher than that of lung cancer patients with normal T_3_ expression [117]. In NTIS patients with both small cell and non-small cell cancer, T_3_ was associated with disease stage and served as a poor prognostic factor [118]. In the human non-small cell cancer cell lines NCI-H522 and NCI-H510A, T_4_ induced a significant increase in proliferating cell nuclear antigen (PCNA) as well as high concentrations of T_3_ [81]. These experiments suggest that high expression of T_4_ in ERα-positive lung cancer cells induces phosphorylation of ERα and activation of ERK1/2, further enhancing PCNA expression and proliferative activity. The collective findings indicate that T_4_ is correlated with lung cancer through inducing hyperthyroidism, which may be a potential risk factor. In an animal model of Lewis lung carcinoma (3LL), cell growth progression is associated with reduced T_4_ and T_3_ levels. Subcutaneous injection with T_4_ induces a hyperthyroid stage and further increase in T_4_ and T_3_ levels associated with primary tumor growth and development of pulmonary metastases 3LL cells. However, the functional effect of T_3_ on 3LL cells is distinct from that of T_4_, which induces significant inhibition of pulmonary metastases and prolongs mouse survival. Additionally, THs (T_4_ and T_3_) treatment promotes cytotoxicity against 3LL cells mediated by alveolar macrophages [119]. Alterations in THs expression influence against primary tumor formation and metastases of lung cancer in the natural host. In lung cancer, THs regulate tumor proliferation and activation of MAPK/ERK1/2 signaling via interactions with TRs on αvβ3 integrin. Around 61% of small-cell lung cancer (SCLC) and 48% of non-SCLC (NSCLC) cases lack TRβ1 expression. While TRβ1 mutations are not commonly observed in human cancers, aberrant TRβ1 through epigenetic regulation has been reported. Neither SCLC nor NSCLC cell lines display somatic mutations of TRβ1. However, methylation of the *TRβ1* promoter has been shown to induce significant loss of *TRβ1* mRNA expression [120].

### 3.4. Brain Tumors

Clinical studies to date have reported extremely low survival rates of patients with glioblastoma multiforme (GBM), the most malignant brain tumor type, with the poorest prognosis. The hypothyroid overall survival rate is longer than that of euthyroid patients [73]. Experiments by another group demonstrated that THs stimulate glioblastoma cell proliferation. Similar to data obtained with other solid tumors, hypothyroidism is associated with improved duration of survival [82,83]. Studies on 230 patients with primary brain tumors showed that 27% had NTIS with high 5-year mortality and short overall survival [121,122]. Additionally, NTIS is a poor prognostic marker for patients undergoing brain tumor surgery. T_4_ can further stimulate glioblastoma growth through interactions with integrin αvβ3 in the cytoplasm [57,82,83,84]. Both T_4_ and T_3_ act as stimulators that induce proliferating cell nuclear antigen (PCNA) accumulation in glioma cells via activation of MAPK/ERK1/2. T_3_ induces PI3K activation, Src kinase and MAPK/ERK1/2 signaling in U-87MG cells. Stimulation of the PI3K pathway by T_3_ promotes TRβ translocation from the cytoplasm to nucleus and activation of *HIF-1α* mRNA transcription [57]. Interestingly, TRα1 and TRα2 expression in human astrocytomas decrease while TRβ1 expression increases with grade of malignancy [85]. Additionally, in the human medulloblastoma cell line, HTB-185, increased TRα2 transcriptional activity is not accompanied by TRβ2 expression [123].

### 3.5. Liver Cancer

Hypothyroidism plays an important role in liver carcinogenesis. A previous study reported hypothyroidism over ten years in female patients with hepatocellular carcinoma, indicating a significant association of hypothyroidism with high risk of human hepatocellular carcinoma (HCC) in females independent of known risk factors [124]. Additional findings support the prevalence of hypothyroidism in HCC patients. The correlation between thyroid function and cancer risk in HCC is related to induction of TSH under the hypothyroid state. In HCC tissue, thyroid stimulating hormone receptor (TSHR) is overexpressed. Therefore, under a hypothyroid state, TSH secretion and synthesis are induced in response to increased binding with highly expressed TSHR and promote HCC progression [125]. A number of studies have demonstrated that T_3_ in HCC downregulates oncogenic CDK2, cyclin E and phospho-Rb, upregulates the tumor suppressor p21 and further inhibits cancer cell proliferation [86,87]. Additionally, T_3_ inhibits cell invasion and metastatic potential via inducing DKK4 expression by reduction of matrix MMP2 and downregulation of the transcription factor ELF2 associated with tumor growth and cell proliferation [88,89]. In HCC tissue and HCC cell lines, both TRα and TRβ display high dominance of truncating and point mutations [126,127,128] Interestingly, *v-erbA* is not only an oncogene but also translates to a mutant form of TRα. Oncogene activity is possibly stimulated in hepatocytes via *v-erbA* dominant-negative activity on T_3_-responsive genes [90]. The *v-erbA* dysregulated T_3_ responsive genes include follistatin, activin βC, thrombomodulin, *Six1*, *Rasgrp3* and *Ndrg2*, which are involved in carcinogenesis leading to HCC development in a transgene mouse model [90]. An earlier study suggested that the *v-erbA* oncogene of avian erythroblastosis virus is devoid of ligand-binding ability. In transgenic mice expressing *v-erbA*, some abnormal physiological phenomena were observed, including breeding disorders, abnormal behavior, reduced adipose tissue, hypothyroidism with inappropriate TSH response and enlarged seminal vesicles. As stated above, earlier experiments supported the development of hepatocellular carcinoma in male animals via *v-erbA* modulation and further identified aberrant (or mutant) TR in correlation with carcinogenesis [129]. On the other hand, T_3_ is reported to be upregulated lipocalin 2 and promote HCC invasiveness by TRs. Additionally, in HCC patient samples, both lipocalin 2 and TRα are overexpressed, which are correlated with cancer grade, stage and survival [91]. T_4_ can activate NF-κB, which is required for THs to induce HCC self-renewal and increase cancer stem-like cells and drug resistance [92]. Finally, similar to other cancer types, T_3_ induces HCC growth by activating ERK1/2/Akt signaling via binding to integrin αvβ3 [93]. In mice, dependent on hyperthyroid conditions and expression of TRα1 in HCC cells, HCC invasion and metastasis are promoted through mediation of MET/FAK pathways. Simultaneously, clinical researchers have further confirmed that in human HCC, tissue samples highly express TRα1 in association with lower survival. To date, no significant correlation between TRβ1 and HCC clinical tissue sample has been established. However, in other HCC specimens, decreased expression of TRs (including TRα1) is reported, supporting a tumor suppressor role [94]. Thus, T_3_/TR may play a dual role (either as an oncogene or tumor suppressor), depending on the molecular background and disease stage. However, this hypothesis remains to be tested.

### 3.6. Colorectal Cancer

Significant clinical research has focused on colon cancer in addition to breast, prostate and lung cancer types. The pathogenesis rate of colon cancer has been shown to be increased by thyrotropin [72]. Data from the Molecular Epidemiology of Colorectal Cancer (MECC) study in Israel indicate that treatment with L-thyroxine for five years can reduce the risk of large bowel cancer [130]. In another study, patients with colorectal cancer displayed reduced T_3_ levels in plasma associated with systemic metastases, implying that inhibition of thyroid hormone signaling further suppresses colorectal cancer invasiveness [131]. This effect may be specifically associated with NTIS. A number of studies propose a possible link between colorectal cancer and thyroid hormone disorders, considering circulation of thyroid hormones, thyrotropin levels and duration of thyroid disease. An earlier investigation showed increased rT_3_ levels in plasma in 24% of colorectal cancer cases and elevated rT_3_/T_3_ ratio in patients with metastatic colorectal cancer [131]. In view of this finding, it is hypothesized that rT_3_ accumulates in metastatic colorectal cancer via deiodinase type 3 (D_3_) activity, D_3_ converts T_3_ to T_2_, thus attenuating T_3_ action locally, and the T_4_/T_3_ ratio is increased, supporting a critical role of thyroid hormone in modulating colorectal cancer metastasis. The mechanism underlying polypoid growth of colorectal cancer remains unclear but may be associated with trophic stimuli, such as thyroid hormones. Colorectal cancer has a high level of integrin αvβ3-expressing tumor vasculature and is associated with significantly lower relapse-free and overall survival, compared to that of patients with low integrin αvβ3 levels in tumor vasculature [132]. Based on these observations, it is proposed that thyroid hormones not only modulate colorectal cancer progression through cell surface integrin αvβ3 but also that integrin αvβ3 acts as a prognostic indicator of colon carcinoma [132]. Colorectal cancer proliferation is induced by T_4_ and thyroxine, both in vitro and in vivo, in a dose-dependent manner in HCT-116, HT-29 and Colo205 cell lines [95,96]. In addition, colorectal cancer progression is promoted through enhancing epithelial cell proliferation in the colon mucosa by T_4_ [133]. Treatment of thyroidectomized rats with T_4_ has been shown to enhance the number of cells per intestinal crypt, suggesting an important regulatory role in intestinal stem-progenitor cell proliferation. However, T_4_ only affects tumor cell proliferative activity and not the histological appearance of colorectal tumors and colon adenocarcinoma type and depth [134]. As stated above, deiodinase type 3 (D_3_) is also involved in colorectal cancer and reported to promote human colon adenoma and adenocarcinoma relative to healthy surrounding mucosa [97]. The β-catenin/T-cell factor complex activated in colorectal cancer stimulates D_3_ expression and T_3_-induced colorectal cancer cell proliferation. However, in both xenograft mice and colon cancer cells, knockdown of β-catenin either led to reduced cell proliferation or enhanced cell differentiation. Overexpression of TRα1 in intestinal cancers promotes hyperproliferation and accelerates the tumorigenic process both in vivo and in vitro [98]. Upon TRα1 overexpression in Apc^+^/1638N mice, tumor appearance, metastasis and intestinal cancer progression are observed. In Apc^+^/1638N mice overexpressing vil-TRα1 (intestinal epithelium-targeted overexpression of TRα1), the tumor development rate is higher than that in control Apc^+^/1638N mice, suggesting that TRα1 is also involved in tumorigenesis via activation of β-catenin [98]. Wild-type TRα1 is beneficial for intestinal regeneration after γ-irradiation-induced DNA damage. However, mutation of TRα1 via irradiation neither induces apoptosis nor reduces cell proliferation. An animal model with the TRα1 mutation showed delayed p53 phosphorylation after induction of the DNA damage response. These experiments indicate that TRα1 plays an important role in intestinal cancer progression via modulation of cell renewal and apoptosis after irradiation treatment and accumulation of DNA damage [135]. Additionally, T_3_ interacts with TRα1 via activation of Frizzled-related protein, sFRP2, which directly modulates β-catenin expression, a major regulator of intestinal cell proliferation [71,136]. Other studies suggest that loss of TRβ1 promotes malignant transformation of human colon tumors. Conversely, overexpression of TRβ1 is associated with reduced invasive activity and consequent biopsies in patients with colorectal cancer [137].

### 3.7. Thyroid Hormone Is Anti-Apoptosis in Cancer Cells

Previous studies reported that thyroid hormone not only play a key role to promote cancer cells proliferation but also induces anti-apoptosis. In breast cancer, the transcription of the pro-apoptotic Bcl-2-associated death promoter (Bad) gene and X-linked inhibitor of apoptosis (XIAP) gene are downregulated by T_3_. Additionally, both T_3_ and T_4_ not only decrease some of the pro-apoptosis markers including caspase-3, Bax but also increase expression of XIAP [63]. The action of T_3_ in the human glioblastoma U-87 MG cells accumulate hypoxia-inducible factor (HIF)-1α mRNA (Figure 5A), which has anti-apoptosis function of gene, via activation of the PI3K pathway by αvβ3 [57]. Under the condition of T_3_, the human kidney showed an increase in the myeloid cell leukemia-1 (MCL-1) transcription activity, which is an anti-apoptotic member of Bcl-2 family, via the presence of TRβ to prevent the formation of channels to release mitochondrial cytochrome c and induce apoptosis [138]. As stated above, we know that most THs promote cancer progression via activation of the signal transduction pathway, one is the MAPK/ERK1/2. Therefore, an ERK1/2 inhibitor, PD98059, which inhibits cascade of MAPK/ERK1/2 signaling transduction and further blocks thyroid hormone-induced cell proliferation [139]. In addition, resveratrol can activate ERK1/2 and induce accumulation of nuclear cyclooxygenase (COX-2) via directing interaction with the integrin αvβ3 receptor to further induce phosphorylation of ERK1/2, which translocates into the cell nucleus and complexes with inducible COX-2 in resveratrol treatment cancer cells. Immediately, the induction of p53-Ser15 phosphorylation can promote downstream gene expression including c-fos, c-jun, and p21 gene expression, and further induce cancer cell apoptosis. However, the action of T_4_ prevents p53-Ser15 phosphorylation and blocks induction cancer cell apoptosis [139]. These results strongly suggest that the thyroid hormone has anti-apoptosis function.

## 4. TH Analogs Exert Anti-Proliferative Effects on Cancer Cells

Cancer cells display enhanced proliferation, reduced apoptosis and increased angiogenesis owing to both genomic and non-genomic actions of THs [140,141,142]. L-thyroxine (T_4_) is the primary thyroid hormone produced by the thyroid gland, which generates T_3_ via outer thyroid hormone ring deiodination at the 5′ position of the diphenyl ether structure of iodothyronines [143]. An alanine side-chain modification of iodothyronine occurs at the cellular level to further generate tetraiodothyroacetic acid (Tetrac) and triiodothyroacetic acid (Triac), both of which have metabolic activities [144]. Accumulating studies have demonstrated the presence of a receptor that can bind thyroid hormone analogs on the plasma membrane of multiple cancer types and rapidly divide endothelial cells [3,51,107], and focused on determining the functions of thyroid hormone analogs. TH derivatives have been effectively utilized to distinguish beneficial actions or deleterious effects of THs for potential therapeutic application [145,146,147,148].

### 4.1. Tetraiodothyroacetic Acid (Tetrac)

Tetraiodothyroacetic acid (Tetrac) is an iodothyronine analog that inhibits T_4_ and T_3_ interactions with the cell surface receptor site of integrin αvβ3 and blocks TH-mediated activation of MAPK/ERK1/2 by principal iodothyronines. Tetrac competes with the binding site on integrin αvβ3, leading to blockage of cancer cell responses to TH. In addition, in the absence of T_4_ and T_3_, Tetrac can influence regulated coherent activities with induction of cell death, modulating the cancer cell survival pathway. Tetrac prevents the activity of T_4_, thus restoring p53-dependent proapoptotic properties of cancer cells [79]. However, in the absence of T_4_, Tetrac does not influence the activity of stilbene resveratrol, which initiates p53-dependent apoptosis through integrin αvβ3 in cancer cells [84]. Additionally, exposure of the estrogen receptor-negative human breast cancer cell line, MDA-MB-231, and medullary thyroid cancer cells, to unmodified Tetrac and nanoparticulate Tetrac (Nanotetrac), which is specifically covalently bound, revealed differential involvement in cancer survival and apoptosis pathways based on effects on gene transcription determined via microarray [149,150]. Tetrac treatment of MDA-MB-231 cells led to both downregulation of the apoptosis inhibitor, XIAP, and increased levels of the angiogenesis inhibitor, thrombospondin 1 (THBS1). On the other hand, Tetrac induced a significant increase in FAF1, CASP8AP2, DFFA, and CASP2 mRNA levels and inhibition of the angiogenic and metastatic effects of vascular endothelial growth factor (VEGFR) and basic fibroblast growth factor (bFGF) in cancer [151,152] (Figure 5B). Moreover, increased levels of CBY1, which inhibits β-catenin function in the nucleus, were observed in the presence of Tetrac [149,150]. Tetrac could directly modulate β-catenin abundance in the cell through downregulation of CTNNA1 and CTNNA2 mRNA (Figure 5B). According to a number of earlier studies, β-catenins are involved in cell-cell adhesion. However, mutation and overexpression of β-catenins occur in various cancer types, including colorectal carcinoma and breast and ovarian cancers [153,154]. In addition, mutation of one of the catenin genes, CTNNA1, which functions as a tumor suppressor in normal conditions, leads to susceptibility to gastrointestinal tract cancers [155], and mutation of the other catenin gene, CTNNA2, is associated with tumor invasiveness [156]. Nanotetrac is excluded from the cell interior, and inhibits cancer cell proliferation induced by thyroid hormone more effectively than unmodified Tetrac [63]. mRNA expression of anti-apoptotic MCL-1 is reduced and apoptosis-promoting CASP2, BCL2L14, and EGFR levels increased by Nanotetrac [63] (Figure 5B). In particular, Nanotetrac downregulates nine genes involved in the cell cycle, one of which is cyclin-dependent kinase gene, along with >20 oncogenes, including Ras family members [63,149]. Nanotetrac also influences tumor invasion, angiogenesis, and metastasis via regulation of matrix metalloproteinase-2 (MMP-2) whose major function is blockage of STAT3 activation by the thyroid hormone [157,158,159] (Figure 5B). The collective studies to date indicate that Tetrac modulates a series of genes involved in cancer cell proliferation, apoptosis and angiogenesis via inhibition of T_4_ and T_3_ actions on integrin αvβ3.

### 4.2. Triiodothyroacetic Acid (Triac)

Tetraiodothyroacetic acid (Tetrac) and 3,3′,5-triiodothyroacetic acid (Triac), acetic acid analogs of THs, have similar structures with thyromimetic activity in the nucleus [160]. Earlier in vitro findings suggest that Triac binding affinity is better than that of T_3_ for both normal TRα1 and TRβ (~three-fold) [161]. Studies on dominant-negative mutants of TRβ in cancer have confirmed that Triac is more powerful than T_3_, i.e., Triac can overcome the dominant-negative effects of TRβ mutations [161]. Triac initiates apoptosis in human ovarian cancer cells via non-genomic actions through effects on integrin αvβ3 and does not influence mitochondria in tumor cells [162]. Nevertheless, the mechanisms by which Triac modulates genomic effects within cancer cells remain to be established. Triac does not interact with genetically modified TRβ. Genetically modified TRβ is trafficked between the cytoplasm and nucleus in cancer cells in the presence or absence of T_3_ and Triac; models can detect endocrine-disrupting chemicals (EDCs) that are a major health concern [163]. Long-term studies using non-cancer cells [164] have shown that Triac binds TR and can be used to clinically treat some forms of thyroid hormone resistance (THS) due to hypersecretion of TSH through elevating transport across the plasma membrane of normal cells [165,166,167,168,169,170,171,172,173,174,175]. However, the majority of findings to date suggest a significant reduction in basal TRH-stimulated TSH levels and serum-free T_4_ and T_3_ levels. In normal and hypothyroid patients, treatment with Triac led to inhibition of TSH synthesis and secretion without inducing changes in peripheral tissue metabolic effects, such as weight, heart rate, reflex time or serum concentrations of cholesterol or triglycerides [175]. Triac modulates normal growth and bone maturation in children with peripheral thyrotoxic features (pituitary RTH (PRTH)) owing to genetic mutation of TRβ (TRβ1) at exon 10 position 1642 (C to A) that generates the amino acid codon change, P453T. TRβ1 mutation led to significantly reduced T_3_ binding affinity. However, upon long-term treatment with Triac, normal heart rate, neurological and clinical signs were maintained in children. These findings suggest that administration of long-term Triac therapy in childhood is safe and efficacious for PRTH [167]. However, no influence on thyroid hormone function after discontinuation of Triac was reported in a proportion of patients, leading to reservations on the specificity of the compound.

## 5. Conclusions

Interactions between nuclear receptors and corresponding ligands that elicit critical molecular pathways have been verified in both normal and cancer cells [4]. This review provides an overview on the individual mechanisms underlying physiological TH/TR-mediated regulation of different cancer types. TRs have been detected in the majority of organs. The mRNAs (THRA and THRB) of TRs generated via alternative splicing produce different isoforms that exert specific physiological effects on the organs [176,177,178]. Additionally, thyroid hormones and TRs play important regulatory roles in physiological processes via actions on intracellular and extracellular proteins and activate various genomic or non-genomic functions including cellular growth, embryonic development, differentiation, metabolism, and proliferation. In normal cells and tissues, both T_4_ and T_3_ modulate skeletal development, bone turnover and maintenance and metabolism of organs [2,143]. T_4_ is a potent thyroid hormone with proliferation-enhancing activity in cancer cells that interacts with αvβ3 integrin, which may serve as a potential target for inhibition of cancer proliferation. T_4_-αvβ3 inhibitors are not particularly efficacious and currently under preclinical analysis for treatment of a few cancer types. However, when disruption of TH signaling induces downstream target gene dysregulation or dysfunction, several organ diseases are initiated. Clinical researchers have suggested that TH/TRs promote proliferation of multiple cancer types including breast cancer, thyroid cancer, lung cancer, brain tumor, liver cancer, and colorectal cancer. In epidemiologic analyses, the risk of ovarian cancer development was shown to be almost double in patients with a history of hyperthyroidism. Moreover, risk was increased about two-fold for patients with pancreatic and prostate cancer with a history of hyperthyroidism. Both hyperthyroidism and hypothyroidism are known to influence the physiological processes underlying cancer. THs stimulate tumor growth and metastasis in vivo while hypothyroidism may be associated with inhibition of tumor proliferation and development. These findings collectively suggest that induction of hyperthyroidism is associated with progression of multiple cancer types. However, cancer progression pathways present considerable variability and complexity, and therefore, therapy utilizing THs/TRs continues to present several unknown challenges. Additionally, THs have not yet been established as a clinical risk factor or prognostic factor for cancer, partly due to their involvement in numerous physiological mechanisms, leading to conflicting experimental data over several years of research. The finding that different TR isoforms are involved in various tumor types and stages of development supports a dual role of TRs in human cancers, either as an oncogene or a tumor suppressor. However, this hypothesis remains to be investigated in detail. Further studies are required to establish the specific functions of TH/TR in cancers of different molecular backgrounds and stages.

## Figures and Tables

**Figure 1 ijms-20-04986-f001:**
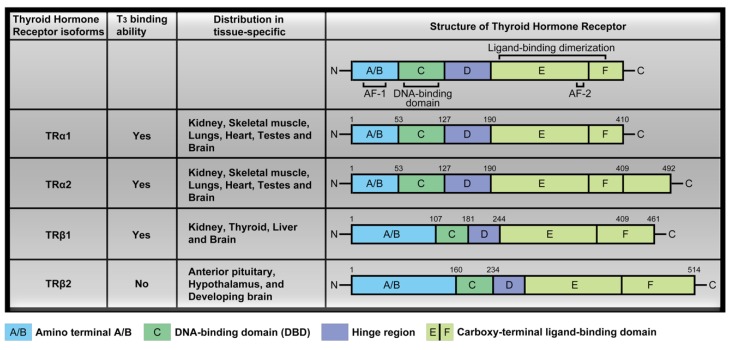
TR isoforms and structure distribution. Thyroid hormone receptors (TR) contain several domains, specifically, amino terminal A/B that may function as a gene enhancer, DNA-binding domain (DBD), hinge region containing the nuclear localization signal and carboxy-terminal ligand-binding domain that binds 3,3,5-Triiodo-L-thyronine (T_3_). The four major TR isoforms, TRα1, TRα2, TRβ1, and TRβ2. TH binding are widely distributed in a tissue-specific manner, for example, TRα1 and TRα2 are expressed in the kidney, skeletal muscle, lungs, heart, and testes, with particularly high levels detected in the brain. Table 1. expression is significant in brain, thyroid, liver, and kidney while the TRβ2 isoform is specifically expressed in the anterior pituitary, hypothalamus, and developing brain.

**Figure 2 ijms-20-04986-f002:**
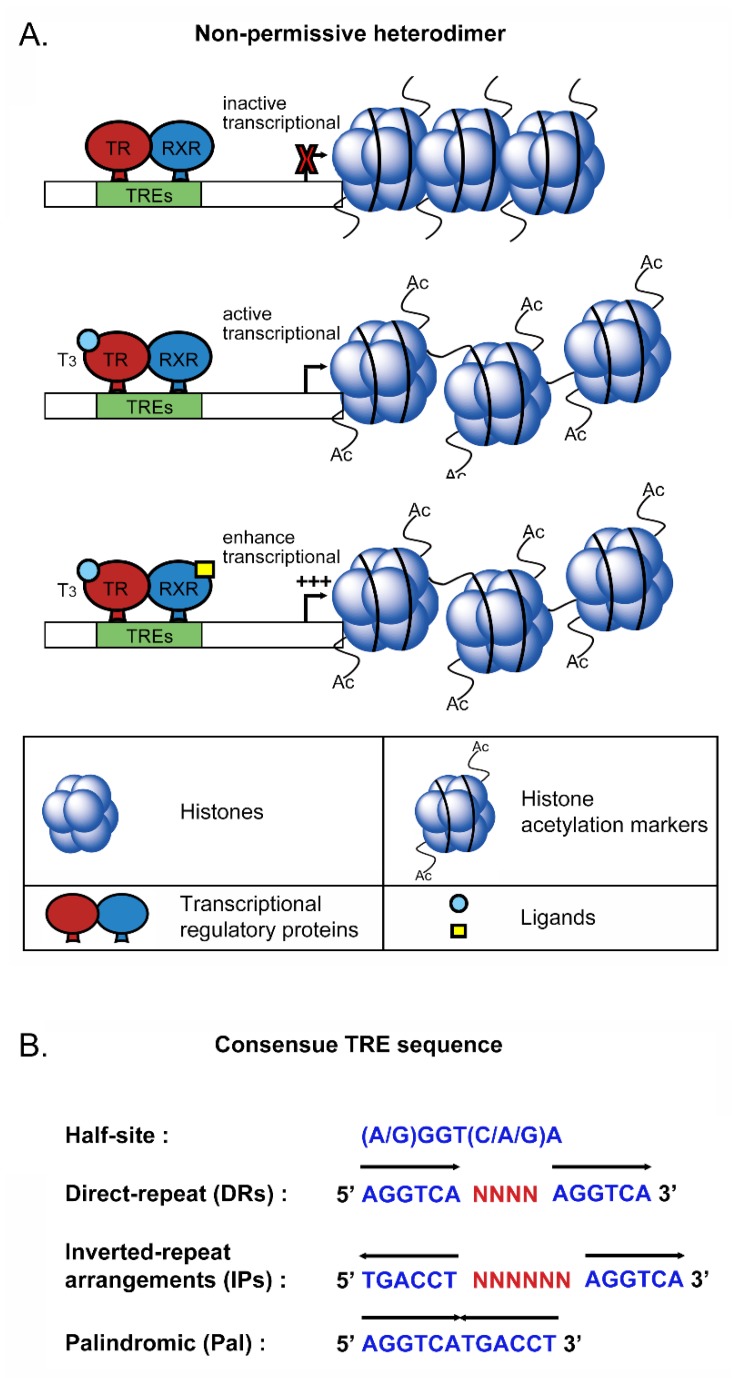
Schematic representation of TRs/RXR non-permissive heterodimers and consensus TRE half-sites. (**A**) TRs/RXR belong to non-permissive heterodimers that can be activated transcriptionally by TRs ligand but not by RXR ligand alone. Moreover, TRs/RXR can enhance the transcriptional response by binding of the RXR ligand and TRs ligand. (**B**) TREs within the promoter region contain individual DNA sequences (A/G)GGT(C/A/G)A, designated “half-sites”, that are recognized by TRs. The half-site sequences in TREs incorporate palindromic (Pal), direct repeat (DR) and inverted repeat arrangements (IP).

**Figure 3 ijms-20-04986-f003:**
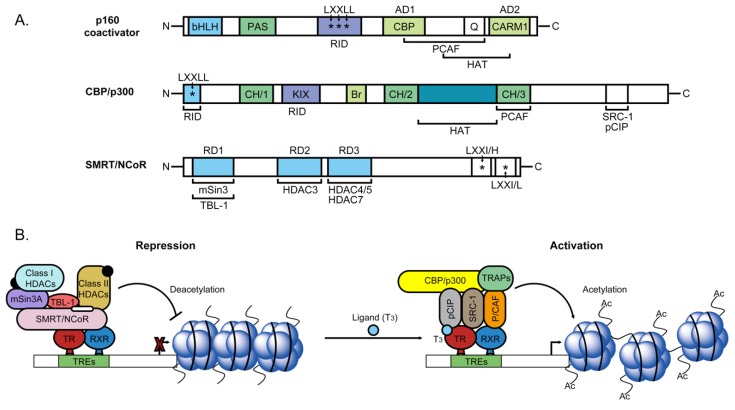
Schematic representation of the structure of receptor coactivators and corepressors and molecular action to regulation of genes transcription activity. (**A**) Biochemical analyses have revealed binding of a series of coactivators proteins to TRs, the most abundant of those proteins such as p160 family, and histone acetyltransferase (HAT), CREB-binding protein (CBP)/p300. Additionally, nuclear receptor corepressor (NCoR) and homolog, silencing mediator of retinoic and thyroid receptor (SMRT), are strongly associated with unliganded TR. (**B**) TRs repress gene expression in a ligand-dependent manner. TRs bind to their respective TREs with retinoid X receptors (RXR). Heterodimers of TR/RXR contain LBDs that interact with T_3_. Additionally, DBD displays high affinity for DNA sequences of TREs. TRs can directly or indirectly associate with different molecular proteins (transcription factors, coactivators, transcription intermediary factors (TIF) and corepressors) to influence downstream target gene expression.

**Figure 4 ijms-20-04986-f004:**
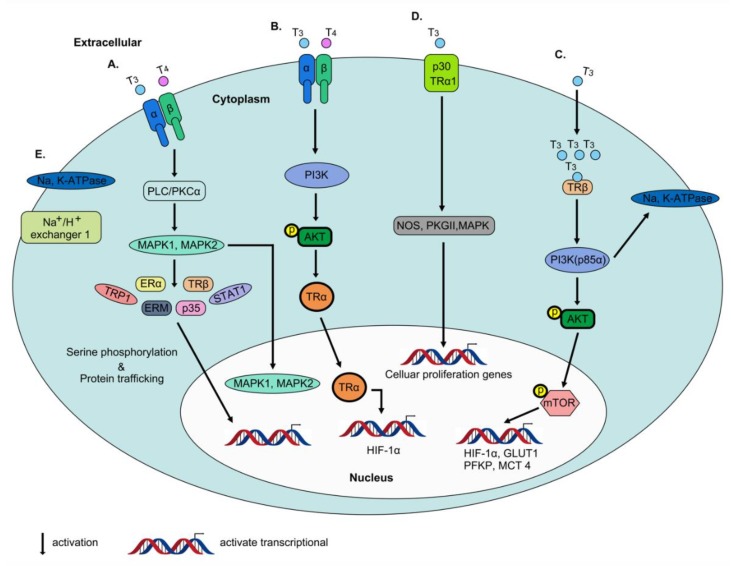
Non-genomic actions of thyroid hormone. THs affect multiple physiological activities within the cell via interactions with integrin αvβ3. (**A**) αvβ3 that binds both T_3_ and T_4_, leading to regulation of MAPK/ERK1/2 via PLC and PKCα, promoting phosphorylation of nucleoproteins TRP1, ERM, STAT1 and p35 and modulation of intracellular protein trafficking of ERα and nuclear uptake of TRβ from the cytoplasm. (**B**) αvβ3 that specifically recognizes T_3_ and activates the PI3K/Akt/PKB pathway via stimulation of Src kinase, leading to direct trafficking of TRα1 from the cytoplasm to the nucleus and transcriptional activity of the target gene, HIF-1α. (**C**) In the cytoplasm, the PI3K pathway is rapidly activated via T_3_ interactions with TRβ1 and initiates downstream target gene transcription including HIF-1α, GLUT1, PFKP and MCT 4. (**D**) At the plasma membrane, T_3_ interacts with p30 TRα1, and binding of PI3K to p85α inactivates transcription of downstream genes. However, the T_3_-liganded TRα1 complex activates a series of signal transduction proteins (PKGII and ERK) and NOS. (**E**) T_3_-liganded TRβ1 stimulates MAPK/ERK 1/2 activity, which activates the sodium proton exchanger (Na^+^/H^+^) in the plasma membrane.

**Figure 5 ijms-20-04986-f005:**
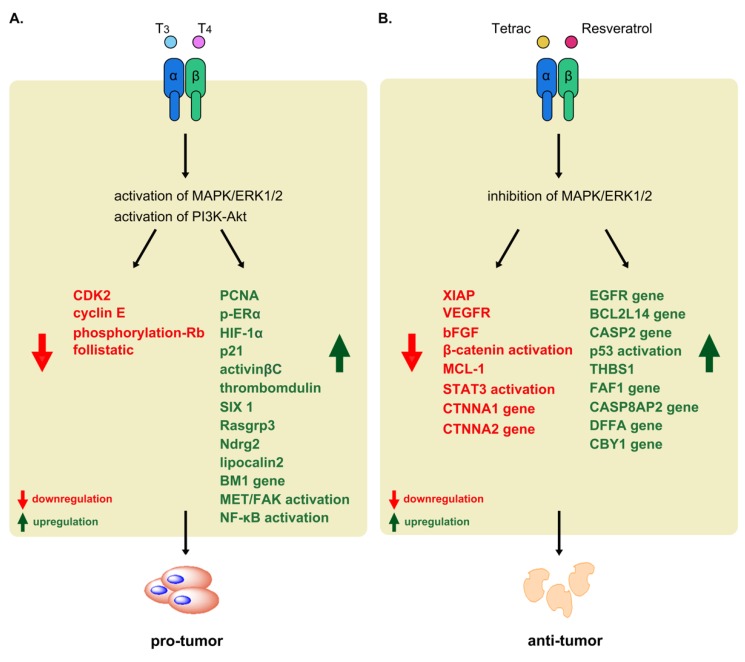
Non-genomic effects of pro-tumorigenic and anti-tumorigenic activities of thyroid hormones in tumor settings. (**A**) αvβ3 that binds both T_3_ and T_4_, leading to activation of MAPK/ERK1/2 and PI3K-Akt, inducing of PCNA, HIF-1α, P21, activinβC, thrombomodulin, SIX1 Rasgrp3, Ndrg2, lipocalin2 and BM1 gene expression to promote phosphorylation of ERα, and activation of MET/FAK, NF-κB, and downregulation of CDK2, cyclin E, phosphorylation of Rb and follistatin, respectively. (**B**) Tetrac can inhibit T_4_ and T_3_ interactions with the cell surface receptor site of integrin αvβ3 and blocks TH-mediated activation of MAPK/ERK1/2 by principal iodothyronines. Tetrac and resveratrol compete with the binding site on integrin αvβ3, leading to blockage of cancer cell responses to TH, leading to upregulation of mRNA of EGFR, BCL2L14, CASP2, FAF1, CASP8AP2, DFFA, CBY1, p53 and TSBH1. Conversely, they downregulate CTNNA1, CTNNA2, XIAP, VEGFR, bFGF mRNA expression as well as inhibit β-catenin and STAT3 activation.

**Table 1 ijms-20-04986-t001:** Summary of the relevant molecular mechanism in different kinds of cancer via THs/TRs.

Cancer Types	Thyroxine	Binding Receptor	Molecular Mechanisms	Physiological Processes	Ref.
Breast cancer	T_3,_ T_4_	αvβ3	activation of MAPK/ERK1/2	proliferation↑	[58,75]
	T_3_	αvβ3	downregulation of SMP30 gene	anti-cancer↑, apoptosis↑	[76]
	T_3_	TRβ	downregulation of T1 gene	proliferation↓	[77]
	T_3_	TRβ	inhibition of STAT5 signaling	development↓	[78]
	T_3_	TRβ	downregulation of β-catenin	prognosis↑	[70]
Thyroid cancer	T_3,_ T_4_	αvβ3	activation of MAPK/ERK1/2	proliferation↑, anti-apoptosis↑	[79]
	T_3_	TRβΔ	activation of PI3K-Akt	metastatic↓, development↑	[80]
	T_3_	TRβ	inhibition of PI3K-Akt	tumor growth↑	[80]
			increase p27	proliferation↓	
			decrease cyclin D		
Lung cancer	T_3,_ T_4_	αvβ3	increase proliferating cell nuclear antigen (PCNA)	proliferation↑	[81]
			induce ERα phosphorylation		
			activation of MAPK/ERK1/2		
Brain tumor	T_3,_ T_4_	αvβ3	increase proliferating cell nuclear antigen (PCNA)	tumor growth↑	[57,82,83,84]
			activation of MAPK/ERK1/2		
	T_3_	TRβ	activation of PI3K-Akt	proliferation↑	[57]
			upregulation of HIF-1α gene		
	T_3_	TRα	expression of TRα1 and TRα2	tumor grade↓, tumor malignancy↓	[85]
Liver cancer	T_3_	TRα, TRβ	downregulation of CDK2, cyclin E, phosphorylation-Rb	proliferation↑	[86,87]
			upregulation of p21		
	T_3,_ T_4_	αvβ3	induction of DKK4	cell invasion↓	[88,89]
			reduction of MMP2	metastatic↓	
			downregulation of ELF2		
	T_3_	TRαΔ	dysregulation of follistatin, activinβC, thrombomodulin, SIX1, Rasgrp3, Ndrg2	development↑, carcinogenesis↑	[90]
	T_3_	TRα	upregulation of lipocalin 2	invasion↑, metastasis↑	[91]
	T_4_	TRα, TRβ	activation of NF-κB	cancer stem like cell↑	[92]
			activation of BM1 gene	drug resistance↑	
	T_3_	αvβ3	activation of ERK1/2/Akt	tumor growth↑	[93]
	T_3,_ T_4_	TRα	activation of MET/FAK	invasion↑, metastasis↑	[94]
Colorectal cancer	T_4_	αvβ3	increase proliferating cell nuclear antigen (PCNA), cyclin D1, c-myc	proliferation↑	[95,96]
	T_4_	TRα	activation of NF-κB	Tumor progression↑, metastasis↑	[97]
	T_3_	TRα1	activation of Frizzled-related protein, sFRP2	proliferation↑	[98]
			modulation of β-catenin

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
