# Peer review of "Molecular Functions of Thyroid Hormone Signaling in Regulation of Cancer Progression and Anti-Apoptosis"

_ijms, 2019, doi:10.3390/ijms20204986_

Round 1

Reviewer 1 Report

In this paper, Yu-Chin Liu and colleagues extensively describe the biological pathways related to Thyroid Hormones in regulating cell biology, in particular in the context of cancer.

The paper is detailed and well written and covers all the aspects of this field. 

I only suggest adding a figure to better illustrate the pro- and anti-tumor effects of thyroid hormones in cancer settings.

Author Response

Response to Reviewer 1 Comments:

Point 1: I only suggest adding a figure to better illustrate the pro- and anti-tumor effects of thyroid hormones in cancer settings.

Authors’ response:  We appreciate the comments of the reviewer. We added the Figure 5. (p.19, lines 590-601, track-change version) discussing thyroid hormones through αvβ3 receptor influence pro-tumor progression by downregulating of CDK2, cyclin E, phosphorylation of Rb and follistatic and upregulating of PCNA, HIF-1α, P21, activinβC, thrombomdulin, SIX1 Rasgrp3, Ndrg2, lipocalin2 and BM1 gene expression to promote phosphorylation of ER?, and activation of MET/FAK, NF-κB, respectively. On the other hand, tetrac and resveratrol competition to thyroid hormones interaction with αvβ3 receptor to further inhibition of downstream signaling transduction. As mentioned above, the cancer cells are blocked the MAPK/ERK1/2 by tetrac and resveratrol, leading to upregulation of mRNA of EGFR, BCL2L14, CASP2, FAF1, CASP8AP2, DFFA, CBY1, p53 and TSBH1. Conversely, they downregulate CTNNA1, CTNNA2, XIAP, VEGFR, bFGF mRNA expression as well as inhibit β-catenin and STAT3 activation.

Reviewer 2 Report

This review is comprehensively well-described.

Major point I think the section 2B&2C are too detailed. They could be focused on the aspects more related to regulation of cancer progression and anti-apoptosis.  

Minor point Table 1 is too small to read.

Author Response

Response to Reviewer 2 Comments:

Point 1: Major point I think the section 2B&2C are too detailed. They could be focused on the aspects more related to regulation of cancer progression and anti-apoptosis.

Authors’ response:  We added a new section suggested by the reviewer related to the regulation of cancer progression and anti-apoptosis. The new section of 3.7. Thyroid hormone is anti-apoptosis in cancer cells (p.14, lines 509-532, track-change version), which discusses that thyroid hormones T3 and T4 in the different types cancer not only mediated αvβ3 receptor downregulate of the pro-apoptotic markers (including Bad, Bax and caspase-3) but also increase of the anti-apoptotic protein expression (including XIPA, MCL-1, HIF-1α). Further, THs promote cancer progression via activation of the signal transduction pathway, such as MAPK/ERK1/2. Thus, we discussed the potential application to use inhibitor and resveratrol blocking induction of cancer cell anti-apoptotic ability by thyroid hormones in this section.

Point 2: Minor point Table 1 is too small to read

Authors’ response:  We revised the tables 1 according to the reviewer’s suggestion (p.14, lines 533, track-change version).